# Long- and short-ranged chiral interactions in DNA-assembled plasmonic chains

Kevin Martens [1], Felix Binkowski [2], Linh Nguyen [1], Li Hu [3], Alexander O. Govorov [4], Sven Burger [2,5] & Tim Liedl [1✉]

Circular dichroism (CD) has long been used to trace chiral molecular states and changes of protein configurations. In recent years, chiral plasmonic nanostructures have shown potential for applications ranging from pathogen sensing to novel optical materials. The plasmonic coupling of the individual elements of such metallic structures is a crucial prerequisite to obtain sizeable CD signals. We here identify and implement various coupling entities—chiral and achiral—to demonstrate chiral transfer over distances close to 100 nm. The coupling is realized by an achiral nanosphere situated between a pair of gold nanorods that are arranged far apart but in a chiral fashion using DNA origami. The transmitter particle causes a strong enhancement of the CD response, the emergence of an additional chiral feature at the resonance frequency of the nanosphere, and a redshift of the longitudinal plasmonic resonance frequency of the nanorods. Matching numerical simulations elucidate the intricate chiral optical fields in complex architectures.

[1] Faculty of Physics, Ludwig-Maximilians-University, Munich, Germany. [2] Zuse Institute Berlin, Berlin, Germany. [3] Chongqing Engineering Laboratory for Detection, Control and Integrated System, Chongqing Technology and Business University, Chongqing, China. [4] Department of Physics and Astronomy, Nanoscale and Quantum Phenomena Institute, Ohio University, Athens, OH, USA. [5] JCMwave GmbH, Berlin, Germany. ✉email: tim.liedl@lmu.de

Chirality describes a geometric feature of structures that do not have any internal planar symmetry. As a consequence, a chiral structure and its mirror image cannot be brought to coincide with each other through the geometrical transformations of rotation and translocation. Such objects of opposite handedness are called enantiomers. They play a decisive role in nature, as a right-handed molecule can have significantly different functions in biological systems as its left-handed counterpart[1]. Along with their distinctive geometrical and thus chemical features, chiral molecules exhibit intricate optical responses upon irradiation with linearly and circularly polarized light. These phenomena are called optical rotatory dispersion (ORD) and circular dichroism (CD), respectively. Amongst others, CD allows to monitor folding processes of proteins[2] and to evaluate the chiral quality of synthetic chemicals[3]. Next to molecular identity and function, chirality enables the creation of designer architectures such as chiral photonic crystals[4,5] and chiral metamaterials[6–8]. Optical activity can already arise from chiral nanoparticles[9–12] (NPs) or assemblies of several achiral NPs into chiral structures through plasmon–plasmon interactions between the surfaces of the NPs[13–18]. CD responses of chiral plasmonic systems show great potential in applications ranging from chiral discrimination of molecules[19] and sensing[20–22], over enantiomer-selective catalysis[23] to circular polarizing devices[7].

Next to CD signals arising from pure organic compounds on the one side and inorganic particles or assemblies on the other side also interactions between these two domains can occur and give rise to strong effects. Particularly, plasmonic surfaces and particles can strongly increase the CD signals of chiral biomolecules in their vicinity, which in turn can enhance the sensitivity of chiroptical detection of biomolecules[24–30]. In such experiments, the strong, plasmon-induced electromagnetic (EM) near-field couples to the chiral near-field of the biomolecules with the latter having its maximum in the UV and extending only a weak tail into the visible spectral range. This type of CD transfer leads to augmentation of the signal strength in the plasmonic window that predominantly occurs in the visible and near infrared (NIR).

DNA nanotechnology[31] and in particular DNA origami[32,33] proved itself to be a powerful tool to implement complex plasmonic particle assemblies with nanometer accuracy[34]. DNA origami structures are formed from a long single-stranded DNA, serving as a scaffold strand, folded into shape by hundreds of synthetic staple oligonucleotides[31]. NPs can be attached by functionalizing them with thiol-modified oligonucleotides that then hybridize at specific positions on the origami structure. These features make DNA origami an ideal platform for nanostructures with tailored optical functionalities[16–18].

DNA origami has been used to achieve chiral nanorod (NR) assemblies in a variety of ways, where rods crossing each other in an X-shape or an L-shape are predominant[35,36]. Switchable variants of these geometries[37–40] further allow for sensitive detection of biomolecules[22]. In all of these assemblies, the distances between the nanoparticles play a crucial role, since plasmon–plasmon interactions are usually limited to the range of a few nanometer[35,36,38–40]. Via sufficiently small gaps, plasmonic energy can traverse efficiently over NP chains[41] and plasmon transfer can even occur between NPs with non-identical resonance frequencies by quasi-occupation of different transfer channels[42]. Transfer of chiral signatures over chains of particles has been predicted theoretically by authors of this study[43], however, experimental prove has been lacking.

Here, we explore this new type of transfer over long distances. In our experiments, plasmon-assisted chiral interactions occur in chiral assemblies of two nanorods arranged with a surface-to-surface distance of over 60 nm. The presence of a third, spherical

transmitter particle in the gap between the rods efficiently couples the near-field of the rods leading to strong signal increase of the longitudinal modes, and the evolution of new CD features in the spectral range of the spherical particle.

## Results and discussion

We designed and synthesized a compound DNA origami platform composed of two individual structures (Fig. 1; see Supplementary Information note S1 for design details). The full structure has an overall length of 100 nm and, by using thiol-DNA functionalization and specific handle strands, accommodates two gold NRs (each 54 nm long and 23 nm wide) at its ends. The rods are designed to have a surface-to-surface distance of 62 nm and they are tilted by 90° in respect to each other when observed along the axis of the origami structure (Fig. 1b). Also in this perspective, each one of the NRs overlap, resulting in an L-shaped object. To serve as a plasmonic transmitter, a 40 nm gold nanosphere (NS) can be attached in between the NRs, spaced 12 nm from each NR (Fig. 1a).

To investigate the effect of this transmitter particle, we synthesized samples with only NRs (NR– –NR) and samples with the center nanosphere present (NR–NS–NR). Transmission electron microscopy (TEM) confirmed the assemblies as designed (Fig. 2a, b). Note that the angular correlation between the particles is lost in the TEM micrographs, as the DNA origami structures—that

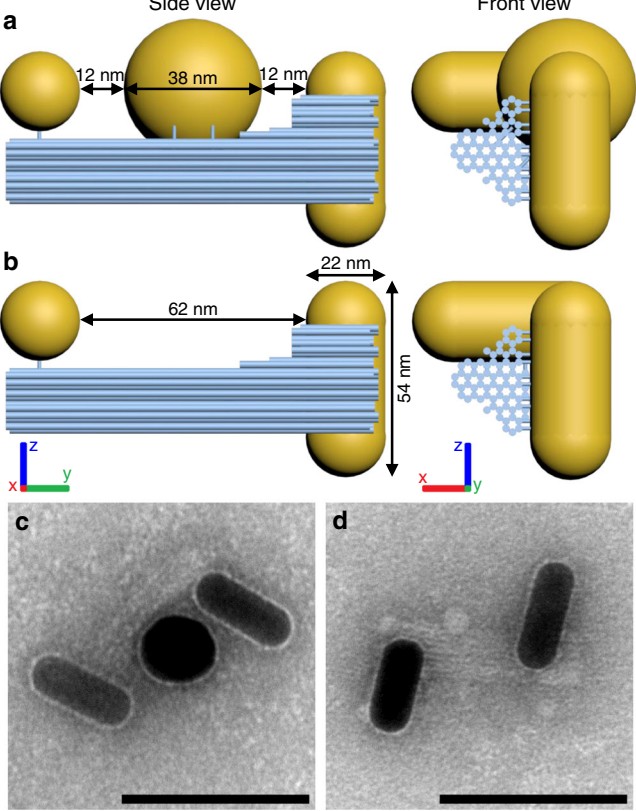

**Fig. 1 Chiral plasmonic transmitter. a** Side view and front view of DNA origami-nanoparticle assemblies in a nanorod–nanosphere–nanorod (NR–NS–NR) arrangement and **b** a nanorod–void–nanorod (NR– –NR) arrangement. The nanorods and the nanosphere are mounted on a DNA origami structure (blue cylinders represent DNA helices) via thiolated DNA strands that are anchored to the origami structure. **c** Transmission electron micrograph of assemblies in the NR–NS–NR arrangement and **d** in the NR– –NR arrangement. Scale bars: 100 nm.

were assembled in solution and will also perform their task in solution—needed to adsorb and dry on the TEM grids before being imaged in vacuum.

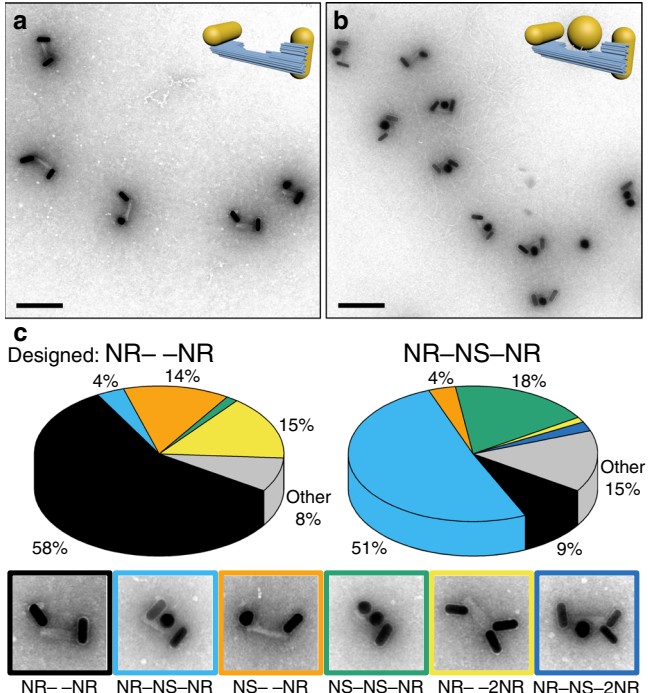

**c** Designed: NR– –NR

4% 14% 15% Other 8% 58%

NR–NS–NR

4% 18% Other 15% 51% 9%

NR– –NR  NR–NS–NR  NS– –NR  NS–NS–NR  NR– –2NR  NR–NS–2NR

**Fig. 2 Assembly yields. a** Electron micrograph of sample NR– –NR, and **b** of sample NR-NS-NR. Scale bar: 200 nm. **c** Distribution of assemblies in the samples NR– –NR and NR–NS–NR. Structures assembled as designed pose the majority with a variety of misassemblies occurring. The colored sections of the pie diagram correspond to the various configurations displayed in the TEM images with colored frames. Approximately 500 individual assemblies were studied (cf. Supplementary Note S3).

A study of 200 assemblies of the sample NR– –NR yielded a majority (58%) to be in the expected arrangement (Fig. 2c and Supplementary Note S3). The most common disarrangement was an extra nanorod attached to the origami structure (NR– –2NR; 15%), which occurs when two NRs attach to the handles designed for a single NR. Another 14% of the structures contained a NS instead of a NR on one side (NS– –NR) due to spurious NSs remaining from NR synthesis. On the other hand, a study of 300 assemblies of the sample NR–NS–NR resulted in 51% well-assembled structures. The most common disarrangements here were structures with a NS instead of a NR attached to one of the ends (NS–NS–NR), contributing to 18% of the sample, as well as assemblies lacking the spherical particle (NR– –NR), contributing to 9% of all assemblies.

The CD measurements were performed with the plasmonic assemblies being dispersed in aqueous solution at concentrations of ~0.1 nM as estimated by the extinction of the NRs longitudinal mode. To compensate for varying sample concentrations, the CD signals were normalized by the same maximum extinction amplitude for each sample. All experimental spectra shown in Fig. 3 thus arise from a multitude (~$10^9$) of individual assemblies, that are present in all possible orientations in the interrogating light path. No contribution of linear dichroism is to be expected in such averaged spectra. Accordingly, the NR– –NR sample shows a typical CD signal for right-handed chiral L-shaped nanostructures with a maximum dip at 648 nm and a peak at 698 nm around the NRs longitudinal plasmon resonance frequency of 676 nm (Fig. 3a, b). In comparison, the NR–NS–NR sample shows the same dip-peak signature but with a 3.5-fold increased amplitude. The peaks are red-shifted to 657 nm (minimum) and 704 nm (maximum), matching a shift in the extinction spectrum for the longitudinal NR peak to a wavelength of 681 nm (Fig. 3b). Additionally, a new CD signature appears in the NR–NS–NR sample around the NS resonance wavelength. At around 512 nm a slight positive deflection can be observed followed by a pronounced negative deflection at 557 nm, with the signal crossing the zero-line close to the plasmonic resonance frequency of the NS at ~530 nm (Fig. 3a, inset).

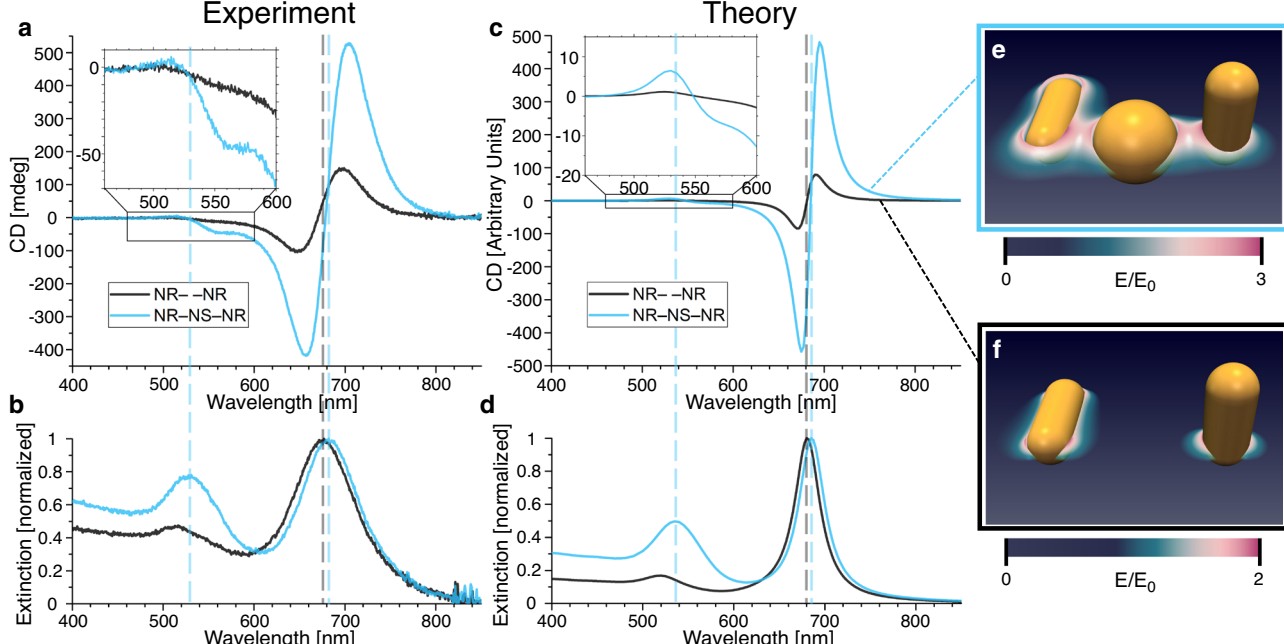

**Fig. 3 CD transfer experiment and theory. a** CD and **b** extinction measurements of samples NR– –NR and NR–NS–NR, normalized by the maximum extinction value. **c** Simulated CD signal and **d** simulated extinction of samples NR– –NR and NR–NS–NR. The signal strength was matched to the experimental amplitude for the CD. **e** Electric near-field intensity around the plasmonic particles shown as a heat map for the arrangement NR–NS–NR and **f** for the arrangement NR– –NR.

Two metallic rods that are arranged in a chiral fashion can be treated as a coupled electron oscillator system. Its electronic response is equivalent to the mechanical Born–Kuhn model leading to analytical solutions in the dipolar limit of large distances[44–47]. In our problem, the optical response arises from the complex interplay of multipolar electromagnetic near fields around two rods and a transmitter particle, which constrained us to compute the CD spectra of the various nanoparticle arrangements by numerically solving the Maxwell's equations. In particular, the gap regions in our trimer structures host so-called hotspots, which represent a challenge for an analytical theory. For solving Maxwell's equations, we use a higher-order finite-element method (FEM), implemented in the solver JCMsuite[48]. The geometry is discretized using a tetrahedral mesh with curvilinear mesh elements along the curved surfaces of the NPs. Employing a conservative setting of the numerical parameters ensured high numerical accuracy (see Supplementary Note S5). We applied tabulated data for the NP material, Au[49], and a constant refractive index of 1.4 for the background material. Circularly polarized plane waves of various wavelengths and incidence directions are used as excitation. The absorbed electromagnetic field energy and the electromagnetic field energy scattered outwards, corresponding to each circularly polarized source term with left-handed circular polarization (LCP) and right-handed circular polarization (RCP), are obtained in post-processes. The extinction is given by the sum of absorption and scattering for both polarization directions, LCP and RCP, and for all six directions of incidence. The CD (g-factor) is given by the difference between absorption and scattering for LCP and for RCP, and is normalized by the extinction maximum of the respective wavelength spectrum (see Supplementary Note S5 for additional details). The simulated spectra are shown in Fig. 3c, d next to heat maps visualizing the coupling via the near-field around the particles (Fig. 3e, f). Strikingly, our models reproduce all the characteristic features of the experimental results.

When compared to the simulations, the recorded data shows many consistencies as well as some noticeable differences. The main and obvious discrepancy between theory and experiment are the broadened dips and peaks and the less pronounced enhancement factor in the experimental data. These differences can be accounted for by the inhomogeneity of the samples, as only small fractions of aggregates can manifest themselves in spectral broadening, which in turn is accompanied by reduced signal increase in case of the transmitter particle present. Notably, due to their achiral nature, the largest part of the disarrangements described above (NS–NS–NR and NS– –NR in the NR–NS–NR sample) cannot be a source of the increased signal strength observed for the NR–NS–NR sample. The main reason for NSs binding to the wrong positions on the DNA structure can be found in the NR purification process. Purifying NRs from NSs, the latter being a side product of our NR synthesis, leaves 12 ± 3% of NSs among the NRs, as deducted from TEM analysis of the sample NR– –NR. Consequently, also the proportion of NS in the sample NR–NS–NR is increased to 41 ± 3%, which is reflected in a higher extinction spectrum at the NS resonance frequency range for both samples.

Interestingly, slight variations in the positioning of the rods in our simulations resulted in strikingly different strengths of the calculated CD responses (Fig. 4). The general behavior can be understood best when thinking about the extreme arrangement of the two nanorods crossing each other in their midpoints. In this case, the structure would depict an achiral + and hence no CD would be observable. In our simulations we moved the NRs outwards, starting from a structure lying between a + and an L-shape (NRP 1). Consequently, only a weak CD response is generated. However, if the ends of the nanorods were moved

outwards (NRP 2–7), the CD response of the longitudinal modes as well as the CD signal around the nanosphere resonance wavelength grew steadily stronger. We hypothesize that such a strong dependency can be a result of the almost matching resonance wavelength between the nanospheres (530 nm) and the transversal plasmon mode of the nanorods (516 nm). Interestingly, we also found the nanorod positions for maximum CD intensity expected to be tolerant to fluctuations of the particle assemblies (Supplementary Note S5).

To summarize, we studied the effect of a spherical transmitter NP in a chiral structure between a NR pair in experiment and simulation. Using the DNA origami technique, we obtained excellent control over the sample geometry and were able to assemble chiral transfer structures with high yields and precision. The chirality response of the NRs can be coupled strongly to a spherical plasmonic particle residing between the rods, where it acts as transmitter. Our matching computational simulations corroborate our understanding of the system and all features of the CD signal can be explained specifically through coupling via the hotspots, and generally through the near-field around the plasmonic particles. The synthesized nanostructures could potentially act as a new type of plasmonic chiral sensors for biomolecules or as transmitter elements for chiral spin locking in optical circuits, where lifting the degeneracy of chiral photons could lead to spin-dependent selection rules of optical transitions.

Supplementary Information containing additional data, protocols and details on calculations is linked to the online version of the paper.

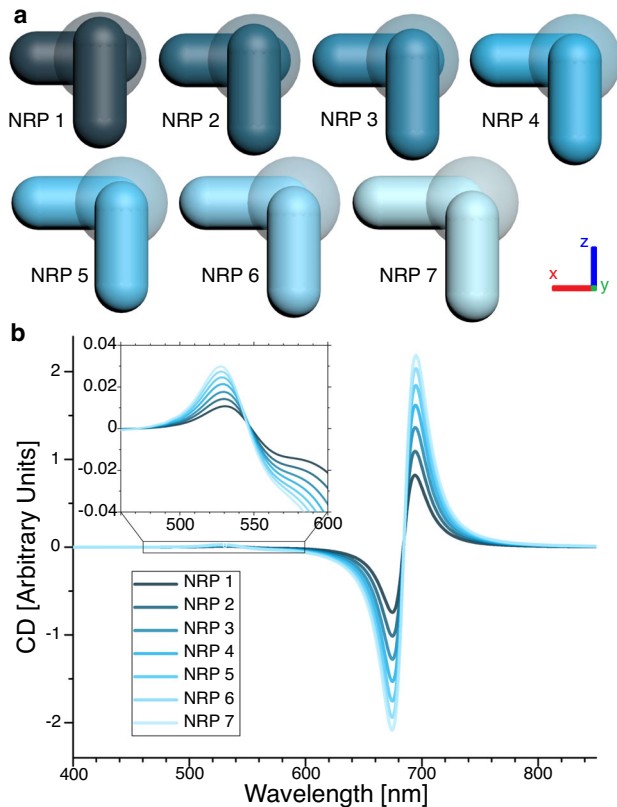

**Fig. 4 CD intensity depends critically on nanoparticle position. a** Slightly varying nanorod positions (NRP) are exemplified for the NR–NS–NR assemblies. In each step, the lower NR is shifted 2 nm in *x*-direction and the upper NR is shifted 2 nm in negative *z*-direction. **b** Simulated CD signals of NR–NS–NR assemblies in NRP 1–7, starting with a weak signal for NRP 1 and ending with a much stronger signal for NRP 7.

## Data availability

All relevant data generated or analysed during this study are included in this published article (and its supplementary information files). TEM images and spectra of replicate samples are available from the corresponding author on request.

## Code availability

A script performing the computations for this work is available as a supplementary data file or from the corresponding author or S.B. on reasonable request. Running the script will require to download and install the JCMsuite FEM solver and to install a license file for it.

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

## Acknowledgements

K.M. and T.L. are grateful for financial support through the ERC Consolidator Grant 818635 DNA Funs and Deutsche Forschungsgemeinschaft (DFG, German Research Foundation) through the SFB1032 (Project A6) and the Cluster of Excellence e-conversion. F.B. and S.B. acknowledge funding by the Deutsche Forschungsgemeinschaft under Germany´s Excellence Strategy—The Berlin Mathematics Research Center MATH+ (EXC-2046/1, project ID: 390685689) and by the German Federal Ministry of Education and Research (BMBF Forschungscampus MODAL, project number 05M20ZBM). The work was supported by the National Natural Science Foundation of China (grant no. 11804035). The ERC Consolidator Grant 818635 DNA Funs also enabled Open Access funding.

## Author contributions

K.M., A.O.G., and T.L. designed the research, K.M. and T.L. designed the nanostructures, K.M. produced the structures and performed the experiments. L.N. synthesized the nanoparticles. F.B., L.H., A.O.G., and S.B. performed theoretical calculations. All authors wrote the manuscript.

## Funding

## Competing interests

The authors declare no competing interests.

**Additional information**

