## [Peer Review File · Nature Communications]

Reviewers' Comments:

Reviewer #1:

Remarks to the Author:

The manuscript describes a very interesting study of hybrid chiral plasmonic – DNA system, and presents a combination of experimental data and numerical modelling. The authors show that the level of optical activity displayed by two nano-rods which adopt a chiral geometry, can be enhanced by the using a spherical nanoparticle to mediate coupling between them.

This is a nice observation, and the synthesis of the hybrid system is impressive. However, the manuscript has weaknesses and should be improved before publications. The two main points that should be addressed are:

1) The combination of two nanorods is paradigmatic of the coupled oscillator (Born-Kuhn) model for optical activity. Indeed the bi-signate line shape of the CD spectrum is signature behaviour for the coupled oscillator model. This discussion, which would support the authors conclusions, and provide a nice narrative framework, is completely missing from the manuscript. This should be addressed.

2) Although the numerical modelling validates the experimental results and show how the coupling fields are enhanced, I think the paper would be strengthened if the level of coupling between the nanorods was quantified. I would suggest the authors apply an analytical model, described in "Microscopic origin of the chiroptical response of optical media" Science Advances 11 Oct 2019: Vol. 5, no. 10, eaav8262 DOI: 10.1126/sciadv.aav8262, to determine coupling constants for their system.

After these changes are made, I would be very happy to recommend publication.

Reviewer #2:

Remarks to the Author:

This manuscript reports the investigation of the effect of a spherical transmitter nanoparticle in a chiral structure between a gold nanorods pair. The investigations are supported both by experimental and theoretical simulation work. The research is very interesting, the structures are really unique and potentially very useful and therefore the manuscript can be published in Nature Communication after some revision. The following points should be clarified.

The CD spectra and interpretation are quite convincing, but the authors should also consider and discuss potential LD effects, which might also occur in these nanostructures. This effect might influence on positioning and shape of the CD peaks.

The matching between theory and experiment is really remarkable. The details of numerical simulations should be included and discussed in the main body of the manuscript as these studies are very important for the interpretations of the phenomena and deserve more attention.

The authors suggest the potential use of the nanostructures as transmitter elements for chiral spin locking in optical circuits. It is not quite clear how this could be achieved and perhaps the authors should elaborate this claim in more details.

Reviewer #3:

Remarks to the Author:

Chiral energy transfer has profound implications in developing new concepts of nanophotonic and information processing devices. In this manuscript, the authors explore the chiral energy transfer between the nanorods at a long distance. They use DNA origami to assemble the chiral nanostructure of two nanorods with a non-chiral nanoparticle as a transmitter. The CD spectra

obtained from both experiment and theory shows significant increase from NR-NR to NR-NS-NR. Also, the CD signal at resonance frequency of nanoparticle transmitter is also observed. The article is clearly written and presents a high-level science. Besides the achievements of this work, some open questions should be noted and discussed, which are listed below.

1. It is theoretically found out that, tiny shift of the nanoparticle position between the two nanorods has great disturbances to the chiral responses, as shown in Figure 4. This influence of the nanoparticle transmitter dictates weak tolerance of the chiral energy transfer property to the structural variation. Such sensitivity could place some difficulties in experimentally fabricating the structures with stable chiral interactions. Is there any chance to achieve robustness of the chiral energy transfer in this system in terms of generating reproducible CD signals? Another problem in this line is to correlate the optical results of sensitive chiral energy transfer with the designed structure, since structural deviations of the nanoparticle assemblies at nanometer scales could occur even with DNA origami method.

2. It was reported before this work that, the far field coupling could be responsible for two twisted nanorods in a large distance. This origin is not discussed in this manuscript. Only the near field coupling mediated by the nanoparticle transmitter is used to account for the chiral interactions between the two distant nanorods. What is the role of the far field coupling play in this optical response?

Point-by-point response

We thank all referees for their positive reception of our work and for very constructive and helpful comments! We changed the manuscript accordingly and added additional data according to your requests.

Reviewer #1 (Remarks to the Author):

The manuscript describes a very interesting study of hybrid chiral plasmonic – DNA system, and presents a combination of experimental data and numerical modelling. The authors show that the level of optical activity displayed by two nano-rods which adopt a chiral geometry, can be enhanced by the using a spherical nanoparticle to mediate coupling between them.

This is a nice observation, and the synthesis of the hybrid system is impressive.

- We thank the reviewer for this positive assessment of our work and for the additional comments answered below, which considerably helped us to improve the manuscript further.

However, the manuscript has weaknesses and should be improved before publications. The two main points that should be addressed are:

1) The combination of two nanorods is paradigmatic of the coupled oscillator (Born-Kuhn) model for optical activity. Indeed the bi-signate line shape of the CD spectrum is signature behaviour for the coupled oscillator model. This discussion, which would support the authors conclusions, and provide a nice narrative framework, is completely missing from the manuscript. This should be addressed.

Thank you for raising this point. The Born-Kuhn model indeed is an excellent framework for two rods arranged in a chiral fashion. We included it into our discussions of the model on page 7.

2) Although the numerical modelling validates the experimental results and show how the coupling fields are enhanced, I think the paper would be strengthened if the level of coupling between the nanorods was quantified. I would suggest the authors apply an analytical model, described in “Microscopic origin of the chiroptical response of optical media” Science Advances 11 Oct 2019: Vol. 5, no. 10, eaav8262 DOI: 10.1126/sciadv.aav8262, to determine coupling constants for their system.

The dipolar model in Science Advances is elegant and certainly very useful for the NR-NR system - we appreciate it fully. However, in our system, the interactions NR-NP-NR are strong and not dipolar (with hot spots contributing significantly to the signal). Therefore, to show our regime, we need the multipolar exposition, and such an analytical model becomes demanding and much less transparent. What would complicate the model further would be that in our case it is necessary to perform summation over all incidence angles (as the nanostructures are dispersed in

solution), which is an extension with regard to the model in this excellent Sci. Adv. paper.

It can be done, but it needs much more time, and is therefore beyond the current time-line. Instead of the analytical model we use numerical modelling. The numerical data are very reliable. We have also validated the used numerical methods by performing simulations of identical setups by two groups independently, by Sven Burger's group (using the FEM solver JCMSuite, as described in the manuscript) and by Sasha Govorov's group who run the simulations with COMSOL and obtained consistent results. To avoid redundancy we do not show the COMSOL results in the manuscript.

Reviewer #2 (Remarks to the Author):

This manuscript reports the investigation of the effect of a spherical transmitter nanoparticle in a chiral structure between a gold nanorods pair. The investigations are supported both by experimental and theoretical simulation work. The research is very interesting, the structures are really unique and potentially very useful and therefore the manuscript can be published in Nature Communication after some revision.

We thank the reviewer for the positive reception of our work.

The CD spectra and interpretation are quite convincing, but the authors should also consider and discuss potential LD effects, which might also occur in these nanostructures. This effect might influence on positioning and shape of the CD peaks.

Thank you for this comment. We added a discussion regarding LD and signal averaging to the main text. Linear dichroism (LD) could indeed occur for individual structures when investigated attached to a substrate, i.e. with restricted (or no) motion. As our samples are dispersed in solution and the multitude of particles can adopt all possible orientations within the interrogating light path that penetrates the solution, usually no LD can be detected. We (and others) have studied this effect in a row of previous works (e.g. references 14-17 and 38-40); Notably, one of our studies deals with chiral DNA origami structures tethered to a surface, discussing various effects of dichroism (reference 18).

The matching between theory and experiment is really remarkable. The details of numerical simulations should be included and discussed in the main body of the manuscript as these studies are very important for the interpretations of the phenomena and deserve more attention.

This is an excellent suggestion. We added a paragraph describing the details of the simulations into the main text (page 7).

The authors suggest the potential use of the nanostructures as transmitter elements for chiral spin locking in optical circuits. It is not quite clear how this could be achieved and perhaps the authors should elaborate this claim in more details.

In analogy to photon transfer along simple photonic waveguides where photons can carry transverse spin angular momentum, which is also locked to the propagation direction, we hypothesize that our devices could lift the spin degeneracy of chiral photons, which could lead to spin-dependent selection rules of optical transitions. We added a sentence to the main body to explain our thoughts.

Reviewer #3 (Remarks to the Author):

Chiral energy transfer has profound implications in developing new concepts of nanophotonic and information processing devices. In this manuscript, the authors explore the chiral energy transfer between the nanorods at a long distance. They use DNA origami to assemble the chiral nanostructure of two nanorods with a non-chiral nanoparticle as a transmitter. The CD spectra obtained from both experiment and theory shows significant increase from NR-NR to NR-NS-NR. Also, the CD signal at resonance frequency of nanoparticle transmitter is also observed. The article is clearly written and presents a high-level science.

Thank you for this positive evaluation and your comments that support our work.

1. It is theoretically found out that, tiny shift of the nanoparticle position between the two nanorods has great disturbances to the chiral responses, as shown in Figure 4. This influence of the nanoparticle transmitter dictates weak tolerance of the chiral energy transfer property to the structural variation. Such sensitivity could place some difficulties in experimentally fabricating the structures with stable chiral interactions. Is there any chance to achieve robustness of the chiral energy transfer in this system in terms of generating reproducible CD signals? Another problem in this line is to correlate the optical results of sensitive chiral energy transfer with the designed structure, since structural deviations of the nanoparticle assemblies at nanometer scales could occur even with DNA origami method.

We thank the reviewer for this very valid question. We agree that it is challenging to build assemblies of nanoparticles with this high accuracy and that this is essential to obtain reliable results. The control over spatial arrangement with DNA origami is indeed exceptional, as others and we have shown (references 34 and Funke & Dietz, Placing molecules with Bohr radius resolution using DNA origami. *Nature Nanotech* 11, 47–52 (2016)). Importantly, we were able to show that AuNPs can be positioned in space with a resolution better than 1 nm (reference 34). Accordingly, we reliably obtain strong CD signals and an increase of signal strength upon insertion of the transmitter particle. Thus, our placement is accurate enough such that g factors can be reached experimentally as high expected from the numerical simulations (Fig. 3).

We assume that the strongest limiting factors are rather the presence of disarrangements as described in Fig. 2c.

Another aspect of this question is, if arrangements could be found that are more tolerant to slight variations of particle positions. We have therefore performed additional simulations for a larger range of relative rod displacements, in order to show that it is possible to find such arrangements with different sensitivities. This worked very well and indeed we found a region of high signal tolerance towards particle fluctuation. These results are now referenced in the main text together with additional simulations shown in the SI (Figure S12). These simulations can also be used to numerically deduce the impact of NP displacement on the CD signal, and corresponding data and discussion has been added to SI note 5.

2. It was reported before this work that, the far field coupling could be responsible for two twisted nanorods in a large distance. This origin is not discussed in this manuscript. Only the near field coupling mediated by the nanoparticle transmitter is used to account for the chiral interactions between the two distant nanorods. What is the role of the far field coupling play in this optical response?

Thank you for pointing this out. The computations include all fields: the near-field and the far-field. The near-field is crucial for strong signals, reflected in the formation of strong hot spots and multipoles from which the chiral signals evolve. Our theoretical methods, based on rigorous simulations of Maxwell's equations, are well established and they also reproduce the far-field spectra very well

Nevertheless, it is indeed interesting to study far field effects in such systems. We therefore added simulations where the distance between the rods was increased up to one wavelength. We found no significant far field effects (SI, Fig. S13) – the near fields dominate at small distances the picture. Why? In our case, the chiral plasmonic complexes are dispersed in solution and randomly oriented. Therefore, we consider just one complex. In a single NR-NS-NR (or NR- -NR) complex, the far-field effect at large distances is very weak since the radiative fields are very small in amplitude. Whereas the near fields in plasmonics are typically very strong and lead to the transfer effects in our case.

Our case of solution is different to the case of planar nanostructures where the far-field interference effects can be noticeable [new SI reference 5; Chiroptical hot spots in twisted nanowire plasmonic oscillators, Yiqiao Tang, Li Sun, and Adam E. Cohen, Appl. Phys. Lett. 102, 043103 (2013)]. In this reference chiroptical effects were observed for the two twisted planes of nanorods separated by a distance about the wavelength; in such a case, the collective response of the 2D arrays leads to interesting chiroptical interference effects.

Reviewers' Comments:

Reviewer #1:

Remarks to the Author:

the authors have revised their manuscript in line with the comments of the reviewers. I'm happy for the paper to be accepted for publication

Reviewer #2:

Remarks to the Author:

The authors have addressed all issues and revised the manuscript appropriately. Therefore, the manuscript is acceptable for publication in the revised form.

Reviewer #3:

Remarks to the Author:

The authors have addressed the questions beautifully in the revised manuscript. I am very glad to recommend publication without further changes.

REVIEWER COMMENTS

Reviewer #1 (Remarks to the Author):

The manuscript describes a very interesting study of hybrid chiral plasmonic – DNA system, and presents a combination of experimental data and numerical modelling. The authors show that the level of optical activity displayed by two nano-rods which adopt a chiral geometry, can be enhanced by the using a spherical nanoparticle to mediate coupling between them.

This is a nice observation, and the synthesis of the hybrid system is impressive. However, the manuscript has weaknesses and should be improved before publications. The two main points that should be addressed are:

1) The combination of two nanorods is paradigmatic of the coupled oscillator (Born-Kuhn) model for optical activity. Indeed the bi-signate line shape of the CD spectrum is signature behaviour for the coupled oscillator model. This discussion, which would support the authors conclusions, and provide a nice narrative framework, is completely missing from the manuscript. This should be addressed.

2) Although the numerical modelling validates the experimental results and show how the coupling fields are enhanced, I think the paper would be strengthened if the level of coupling between the nanorods was quantified. I would suggest the authors apply an analytical model, described in “Microscopic origin of the chiroptical response of optical media” Science Advances 11 Oct 2019: Vol. 5, no. 10, eaav8262 DOI: 10.1126/sciadv.aav8262, to determine coupling constants for their system.

After these changes are made, I would be very happy to recommend publication.

Reviewer #2 (Remarks to the Author):

This manuscript reports the investigation of the effect of a spherical transmitter nanoparticle in a chiral structure between a gold nanorods pair. The investigations are supported both by experimental and theoretical simulation work. The research is very interesting, the structures are really unique and potentially very useful and therefore the manuscript can be published in Nature Communication after some revision. The following points should be clarified.

The CD spectra and interpretation are quite convincing, but the authors should also consider and discuss potential LD effects, which might also occur in these nanostructures. This effect might influence on positioning and shape of the CD peaks.

The matching between theory and experiment is really remarkable. The details of numerical simulations should be included and discussed in the main body of the manuscript as these studies are very important for the interpretations of the phenomena and deserve more attention.

The authors suggest the potential use of the nanostructures as transmitter elements for chiral spin locking in optical circuits. It is not quite clear how this could be achieved and perhaps the authors should elaborate this claim in more details.

Reviewer #3 (Remarks to the Author):

Chiral energy transfer has profound implications in developing new concepts of nanophotonic and information processing devices. In this manuscript, the authors explore the chiral energy transfer between the nanorods at a long distance. They use DNA origami to assemble the chiral nanostructure of two nanorods with a non-chiral nanoparticle as a transmitter. The CD spectra obtained from both experiment and theory shows significant increase from NR-NR to NR-NS-NR. Also, the CD signal at resonance frequency of nanoparticle transmitter is also observed. The article is clearly written and presents a high-level science. Besides the achievements of this work, some open questions should be noted and discussed, which are listed below.

1. It is theoretically found out that, tiny shift of the nanoparticle position between the two nanorods has great disturbances to the chiral responses, as shown in Figure 4. This influence of the nanoparticle transmitter dictates weak tolerance of the chiral energy transfer property to the structural variation. Such sensitivity could place some difficulties in experimentally fabricating the structures with stable chiral interactions. Is there any chance to achieve robustness of the chiral energy transfer in this system in terms of generating reproducible CD signals? Another problem in this line is to correlate the optical results of sensitive chiral energy transfer with the designed structure, since structural deviations of the nanoparticle assemblies at nanometer scales could occur even with DNA origami method.

2. It was reported before this work that, the far field coupling could be responsible for two twisted nanorods in a large distance. This origin is not discussed in this manuscript. Only the near field coupling mediated by the nanoparticle transmitter is used to account for the chiral interactions between the two distant nanorods. What is the role of the far field coupling play in this optical response?

Point-by-point response

We thank all referees for their positive reception of our work and for very constructive and helpful comments! We changed the manuscript accordingly and added additional data according to your requests.

Reviewer #1 (Remarks to the Author):

The manuscript describes a very interesting study of hybrid chiral plasmonic – DNA system, and presents a combination of experimental data and numerical modelling. The authors show that the level of optical activity displayed by two nano-rods which adopt a chiral geometry, can be enhanced by the using a spherical nanoparticle to mediate coupling between them.

This is a nice observation, and the synthesis of the hybrid system is impressive.

- We thank the reviewer for this positive assessment of our work and for the additional comments answered below, which considerably helped us to improve the manuscript further.

However, the manuscript has weaknesses and should be improved before publications. The two main points that should be addressed are:

1) The combination of two nanorods is paradigmatic of the coupled oscillator (Born-Kuhn) model for optical activity. Indeed the bi-signate line shape of the CD spectrum is signature behaviour for the coupled oscillator model. This discussion, which would support the authors conclusions, and provide a nice narrative framework, is completely missing from the manuscript. This should be addressed.

Thank you for raising this point. The Born-Kuhn model indeed is an excellent framework for two rods arranged in a chiral fashion. We included it into our discussions of the model on page 7.

2) Although the numerical modelling validates the experimental results and show how the coupling fields are enhanced, I think the paper would be strengthened if the level of coupling between the nanorods was quantified. I would suggest the authors apply an analytical model, described in “Microscopic origin of the chiroptical response of optical media” Science Advances 11 Oct 2019: Vol. 5, no. 10, eaav8262 DOI: 10.1126/sciadv.aav8262, to determine coupling constants for their system.

The dipolar model in Science Advances is elegant and certainly very useful for the NR-NR system - we appreciate it fully. However, in our system, the interactions NR-NP-NR are strong and not dipolar (with hot spots contributing significantly to the signal). Therefore, to show our regime, we need the multipolar exposition, and such an analytical model becomes demanding and much less transparent. What would complicate the model further would be that in our case it is necessary to perform summation over all incidence angles (as the nanostructures are dispersed in

solution), which is an extension with regard to the model in this excellent Sci. Adv. paper.

It can be done, but it needs much more time, and is therefore beyond the current time-line. Instead of the analytical model we use numerical modelling. The numerical data are very reliable. We have also validated the used numerical methods by performing simulations of identical setups by two groups independently, by Sven Burger's group (using the FEM solver JCMSuite, as described in the manuscript) and by Sasha Govorov's group who run the simulations with COMSOL and obtained consistent results. To avoid redundancy we do not show the COMSOL results in the manuscript.

Reviewer #2 (Remarks to the Author):

This manuscript reports the investigation of the effect of a spherical transmitter nanoparticle in a chiral structure between a gold nanorods pair. The investigations are supported both by experimental and theoretical simulation work. The research is very interesting, the structures are really unique and potentially very useful and therefore the manuscript can be published in Nature Communication after some revision.

We thank the reviewer for the positive reception of our work.

The CD spectra and interpretation are quite convincing, but the authors should also consider and discuss potential LD effects, which might also occur in these nanostructures. This effect might influence on positioning and shape of the CD peaks.

Thank you for this comment. We added a discussion regarding LD and signal averaging to the main text. Linear dichroism (LD) could indeed occur for individual structures when investigated attached to a substrate, i.e. with restricted (or no) motion. As our samples are dispersed in solution and the multitude of particles can adopt all possible orientations within the interrogating light path that penetrates the solution, usually no LD can be detected. We (and others) have studied this effect in a row of previous works (e.g. references 14-17 and 38-40); Notably, one of our studies deals with chiral DNA origami structures tethered to a surface, discussing various effects of dichroism (reference 18).

The matching between theory and experiment is really remarkable. The details of numerical simulations should be included and discussed in the main body of the manuscript as these studies are very important for the interpretations of the phenomena and deserve more attention.

This is an excellent suggestion. We added a paragraph describing the details of the simulations into the main text (page 7).

The authors suggest the potential use of the nanostructures as transmitter elements for chiral spin locking in optical circuits. It is not quite clear how this could be achieved and perhaps the authors should elaborate this claim in more details.

In analogy to photon transfer along simple photonic waveguides where photons can carry transverse spin angular momentum, which is also locked to the propagation direction, we hypothesize that our devices could lift the spin degeneracy of chiral photons, which could lead to spin-dependent selection rules of optical transitions. We added a sentence to the main body to explain our thoughts.

Reviewer #3 (Remarks to the Author):

Chiral energy transfer has profound implications in developing new concepts of nanophotonic and information processing devices. In this manuscript, the authors explore the chiral energy transfer between the nanorods at a long distance. They use DNA origami to assemble the chiral nanostructure of two nanorods with a non-chiral nanoparticle as a transmitter. The CD spectra obtained from both experiment and theory shows significant increase from NR-NR to NR-NS-NR. Also, the CD signal at resonance frequency of nanoparticle transmitter is also observed. The article is clearly written and presents a high-level science.

Thank you for this positive evaluation and your comments that support our work.

1. It is theoretically found out that, tiny shift of the nanoparticle position between the two nanorods has great disturbances to the chiral responses, as shown in Figure 4. This influence of the nanoparticle transmitter dictates weak tolerance of the chiral energy transfer property to the structural variation. Such sensitivity could place some difficulties in experimentally fabricating the structures with stable chiral interactions. Is there any chance to achieve robustness of the chiral energy transfer in this system in terms of generating reproducible CD signals? Another problem in this line is to correlate the optical results of sensitive chiral energy transfer with the designed structure, since structural deviations of the nanoparticle assemblies at nanometer scales could occur even with DNA origami method.

We thank the reviewer for this very valid question. We agree that it is challenging to build assemblies of nanoparticles with this high accuracy and that this is essential to obtain reliable results. The control over spatial arrangement with DNA origami is indeed exceptional, as others and we have shown (references 34 and Funke & Dietz, Placing molecules with Bohr radius resolution using DNA origami. *Nature Nanotech* 11, 47–52 (2016)). Importantly, we were able to show that AuNPs can be positioned in space with a resolution better than 1 nm (reference 34). Accordingly, we reliably obtain strong CD signals and an increase of signal strength upon insertion of the transmitter particle. Thus, our placement is accurate enough such that g factors can be reached experimentally as high expected from the numerical simulations (Fig. 3).

We assume that the strongest limiting factors are rather the presence of disarrangements as described in Fig. 2c.

Another aspect of this question is, if arrangements could be found that are more tolerant to slight variations of particle positions. We have therefore performed additional simulations for a larger range of relative rod displacements, in order to show that it is possible to find such arrangements with different sensitivities. This worked very well and indeed we found a region of high signal tolerance towards particle fluctuation. These results are now referenced in the main text together with additional simulations shown in the SI (Figure S12). These simulations can also be used to numerically deduce the impact of NP displacement on the CD signal, and corresponding data and discussion has been added to SI note 5.

2. It was reported before this work that, the far field coupling could be responsible for two twisted nanorods in a large distance. This origin is not discussed in this manuscript. Only the near field coupling mediated by the nanoparticle transmitter is used to account for the chiral interactions between the two distant nanorods. What is the role of the far field coupling play in this optical response?

Thank you for pointing this out. The computations include all fields: the near-field and the far-field. The near-field is crucial for strong signals, reflected in the formation of strong hot spots and multipoles from which the chiral signals evolve. Our theoretical methods, based on rigorous simulations of Maxwell's equations, are well established and they also reproduce the far-field spectra very well

Nevertheless, it is indeed interesting to study far field effects in such systems. We therefore added simulations where the distance between the rods was increased up to one wavelength. We found no significant far field effects (SI, Fig. S13) – the near fields dominate at small distances the picture. Why? In our case, the chiral plasmonic complexes are dispersed in solution and randomly oriented. Therefore, we consider just one complex. In a single NR-NS-NR (or NR- -NR) complex, the far-field effect at large distances is very weak since the radiative fields are very small in amplitude. Whereas the near fields in plasmonics are typically very strong and lead to the transfer effects in our case.

Our case of solution is different to the case of planar nanostructures where the far-field interference effects can be noticeable [new SI reference 5; Chiroptical hot spots in twisted nanowire plasmonic oscillators, Yiqiao Tang, Li Sun, and Adam E. Cohen, Appl. Phys. Lett. 102, 043103 (2013)]. In this reference chiroptical effects were observed for the two twisted planes of nanorods separated by a distance about the wavelength; in such a case, the collective response of the 2D arrays leads to interesting chiroptical interference effects.

REVIEWERS' COMMENTS

Reviewer #1 (Remarks to the Author):

the authors have revised their manuscript in line with the comments of the reviewers. I'm happy for the paper to be accepted for publication

Reviewer #2 (Remarks to the Author):

The authors have addressed all issues and revised the manuscript appropriately. Therefore, the manuscript is acceptable for publication in the revised form.

Reviewer #3 (Remarks to the Author):

The authors have addressed the questions beautifully in the revised manuscript. I am very glad to recommend publication without further changes.